# Bone Turnover Markers, n-Terminal Propeptide of Type I Procollagen and Tartrate-Resistant Acid Phosphatase Type 5b, for Predicting Castration Resistance in Prostate Cancer

**DOI:** 10.3390/biomedicines12020292

**Published:** 2024-01-26

**Authors:** Hiroshi Kano, Kouji Izumi, Ryunosuke Nakagawa, Ren Toriumi, Shuhei Aoyama, Taiki Kamijima, Tomoyuki Makino, Renato Naito, Hiroaki Iwamoto, Hiroshi Yaegashi, Shohei Kawaguchi, Kazuyoshi Shigehara, Takahiro Nohara, Atsushi Mizokami

**Affiliations:** Department of Integrative Cancer Therapy and Urology, Kanazawa University Graduate School of Medical Science, 13-1 Takara-machi, Kanazawa 920-8640, Ishikawa, Japan; kanazawa_iimati@yahoo.co.jp (H.K.); r_a_rhero0226southern@yahoo.co.jp (R.N.); rentoriumi59@gmail.com (R.T.); a_shuhei_0151@yahoo.co.jp (S.A.); kamiji0029@yahoo.co.jp (T.K.); thealfuu@yahoo.co.jp (R.N.); hiroaki017@yahoo.co.jp (H.I.); hyae2002jp@yahoo.co.jp (H.Y.); shohei_k2001@yahoo.co.jp (S.K.); kshigehara0415@yahoo.co.jp (K.S.); t_nohara704@yahoo.co.jp (T.N.); mizokami@med.kanazawa-u.ac.jp (A.M.)

**Keywords:** P1NP, TRACP-5b, castration-resistant prostate cancer

## Abstract

Bone is a common site of prostate cancer metastasis. Bone turnover markers n-terminal propeptide of type I procollagen (P1NP) and tartrate-resistant acid phosphatase type 5b (TRACP-5b) are highly sensitive to bone remodeling activity. However, their prognostic significance as markers of prostate cancer is unknown. This study retrospectively examined the usefulness of P1NP and TRACP-5b as prognostic biomarkers. Castration-resistant prostate cancer recurrence-free survival (CFS) was estimated using the Kaplan–Meier method. A predictive model for CFS was constructed using multivariate analysis. This study enrolled 255 patients diagnosed with prostate cancer at Kanazawa University Hospital. The median follow-up was 115.1 months. Patients with both high serum P1NP and TRACP-5b levels, defined as having a poor bone turnover category (BTC), had significantly shorter CFS. Multivariate analysis identified Gleason score, metastasis, and BTC poor as predictors for castration resistance in prostate cancer. Using these three factors, a prognostic model was established, categorizing patients into low-risk (no or one factor) and high-risk (two or three factors) groups. In the low-risk group, the median CFS was not reached, contrasting with 19.1 months in the high-risk group (hazard ratio, 32.23, *p* < 0.001). Combining P1NP and TRACP-5b may better predict castration resistance.

## 1. Introduction

Prostate cancer (PC) is the most common malignancy in men [1]. It is initially responsive to androgen deprivation therapy (ADT) and eventually develops into castration-resistant PC (CRPC) in most cases and ultimately death [2]. A concerted effort has been made to explore new biomarkers to enhance diagnostic and prognostic strategies for PC [3,4]. Bone is a major visible metastatic site and is a potential invisible micrometastatic site in PC; thus, bone turnover markers have been reported as prognostic factors for PC and provide valuable information regarding the progression of bone metastasis and bone health under treatment of bone modifying agents, garnering considerable attention [5,6,7].

In adults, bone mass is maintained by continuous bone remodeling, i.e., a balance between osteoblast-mediated bone formation and osteoclast-mediated bone resorption [7]. Osteoblasts are of mesenchymal origin and contribute to bone formation. Osteoblasts first synthesize procollagen, which produces type 1 collagen and n-terminal propeptide of type I procollagen (P1NP)/n-terminal propeptide of type I procollagen (P1CP) propeptide when cleaved at the N- and C-termini. These fragments are released into the blood and undergo hepatic clearance [6]. Then, osteoblasts secrete bone-specific alkaline phosphatase, which hydrolyzes pyrophosphate, a physiological inhibitor of bone matrix maturation, and releases inorganic phosphate P1NP, an indicator of bone formation, specifically early in the process because it is elevated early in bone formation [8]. Osteoclasts are multinucleated bone-resorbing cells derived from the monocyte/macrophage lineage. Osteoclast erosive activity is based on the secretion of H+ ions and lytic enzymes such as proteases and tartrate-resistant acid phosphatase type 5b (TRACP-5b) [6]. TRACP5b is suitable for primary use in assessing osteoclast activity because diurnal variations, food intake, or renal dysfunction do not affect serum concentrations [9]. P1NP has the advantage of fluctuating early in the process, and TRACP-5b, which is not subject to diurnal or external fluctuations, is expected to find new clinical applications.

P1NP and TRACP-5b were previously explored for their associations with bone turnovers and showed their potential as active indicators of disease progression and survival outcomes in PC [6,10]. However, their relevance to clinical parameters conventionally used in PC and clinical outcomes such as cancer progression and patient survival remains unclear. In this study, their correlation with established clinical parameters, including the Gleason score (GS) and TNM staging, was retrospectively analyzed, and their potential as predictors for the development of CRPC was evaluated.

## 2. Materials and Methods

This study enrolled patients who underwent prostate biopsy at Kanazawa University Hospital between 2007 and 2013. Patient characteristics and survival data were obtained from their charts. Patients diagnosed with malignancy were enrolled, whereas those who did not receive a cancer diagnosis among those undergoing biopsy were excluded. All patients who were able to undergo prostate biopsy were included; no other exclusion criteria were established.

A 10-core needle transrectal ultrasound-guided biopsy of the prostate gland, using an 18 G needle, was conducted. Samples were procured from the apex, middle, base, and two outer lateral areas of bilateral peripheral zones. Magnetic resonance imaging (MRI), computed tomography (CT), and bone scintigraphy were used for the TNM staging of PC. The clinical stage was determined based on the 2017 TNM Classification of Malignancies, 8th Edition.

The treatment strategy and imaging intervals following PC diagnosis were entirely at the discretion of each attending physician. The chosen treatment strategy encompassed the continuous or temporary use of ADT with radiotherapy, radical prostatectomy, and active surveillance, or their individual modalities. ADT options comprised surgical castration, monotherapy involving luteinizing hormone-releasing hormone (LH–RH) analogs or antagonists, and combination therapy using anti-androgens and LH-RH analogs or antagonists. If PSA recurrence was observed after local therapy, ADT treatment was performed. Once PC became CRPC, antagonists, androgen receptor signaling target therapy, chemotherapy, and Radium223 were used.

The levels of P1NP and TRACP-5b were measured using blood samples taken simultaneously at the time of prostate biopsy. Serum values of each biomarker were measured using commercially available kits according to the suppliers’ instruction manuals: P1NP (ORION Diagnostica, Espoo, Finland) and TRACP-5b (Nittobo Medical, Tokyo, Japan). As per the accompanying text of the measurement kit: for the reference value of P1NP, measurements were obtained from 131 healthy men (aged 20–80 years), yielding a mean value of 39.8 μg/L. The established reference range is 19.0–83.5 μg/L. For the reference values of TRACP-5b, measurements were derived from 309 healthy men (aged 25–82 years), and the established reference range was 170–590 mU/dL.

We defined PSA failure after ADT as an elevated PSA level of at least 2.0 ng/mL and a 25% increase from the nadir, confirmed by a second PSA test at least four weeks later. The diagnosis of castration-resistant prostate cancer (CRPC) was made when the above criteria were met. CRPC-free survival (CFS), cancer-specific survival (CSS), and overall survival (OS) were evaluated from the diagnosis of PC. Follow-up was terminated in March 2023.

This study was approved by the Institutional Review Board of Kanazawa University Hospital (2013-064).

The association between P1NP and TRACP-5b levels and GS and TNM stage was checked for equal variance with Fisher’s exact test using paired and unpaired Student’s *t*-tests. Correlation coefficients between P1NP and TRACP-5b and age and PSA were calculated using single linear regression analysis. Cutoff values for P1NP and TRACP-5b were calculated with receiver operating characteristic (ROC) curve analysis using the area under the curve (AUC). CFS, CSS, and OS were estimated using the Kaplan–Meier method, and differences were compared using log-rank tests. The Cox proportional-hazards model was used for univariate and multivariate analyses. Multivariate analysis included all factors that were significantly different in the univariate analysis. Statistical analyses were performed using IBM SPSS Statistics version 25.0 (IBM Corp., Armonk, NY, USA) and Prism v.9.4.0 (GraphPad, San Diego, CA, USA). Statistical significance was indicated in all analyses by a *p*-value of <0.05.

## 3. Results

### 3.1. Patient Characteristics

The patient background is shown in Table 1. Of the 379 patients who underwent prostate biopsy at Kanazawa University Hospital, 255 patients who were diagnosed with PC were enrolled in this study. The median follow-up period was 115.1 months. The median age at the time of PC diagnosis was 69 (range, 46–89) years, with a median PSA level of 10.6 (range, 1.5–16,701) ng/mL. Among them, 98 patients (38%) had a GS of 8–10, 62 (24%) exhibited clinical T stage of ≥3, 27 (11%) presented with lymph node metastases, and 25 (10%) had distant metastases. The metastatic cases were 0 for M1a, 16 for M1b, and nine for M1c. The median P1NP and TRACP-5b levels were 37.9 (range, 7.2–511) μg/L and 273 mU/dL (range, 98–1140 mU/L), respectively.

At the time of analysis, 26 patients had progressed to CRPC. Three patients received radiotherapy to the prostate area as local therapy, whereas the remaining twenty-three received treatment with only ADT. Among the total of 49 recorded fatalities, 17 were directly linked to PC. Additional mortalities included 12 cases of diverse cancers—four cases of lung cancer, four of liver cancer, four of pancreatic cancer, and one of bile duct cancer. Furthermore, eight deaths were caused by pneumonia, one by myocardial infarction, one by traffic accident, and 10 deaths had unknown causes.

### 3.2. Relevance to Clinical Parameters

The levels of both P1NP and TRACP-5b were significantly high in patients with GS of 8–10 compared with those with scores of 6 and 7 (*p* < 0.001, Figure 1A,E). Furthermore, in TNM staging, notably higher levels of P1NP and TRACP-5b were observed in patients with T3 and T4 than in those with T1 or T2 (*p* < 0.001, Figure 1A,F). A similar pattern emerged in cases marked by N and M classification (*p* < 0.001 and *p* < 0.001, respectively, Figure 1C,D,G,H). No correlation was observed between P1NP and age (R^2^ = 0.007, *p* = 0.180), nor between TRACP-5b and age (R^2^ = 0.030, *p* = 0.005). Similarly, no correlation was found between P1NP and PSA (R^2^ = 0.027, *p* = 0.009) nor between TRACP-5b and PSA (R^2^ = 0.014, *p* = 0.057).

Subsequently, the prognostic utility of P1NP and TRACP-5b was investigated. ROC curves were constructed to ascertain the optimal cutoff values. The cutoff values for CFS were 55.8 μg/L for P1NP (sensitivity, 73.1%; specificity, 87.8%; and AUC, 0.85) and 400 mU/dL for TRACP-5b (sensitivity, 57.7%; specificity, 87.8%; and AUC, 0.78) (Figure 2). Using the abovementioned cutoff values for patient stratification, the Kaplan–Meier survival curves for CFS based on P1NP and TRACP-5b are presented in Figure 3. Patients with P1NP levels ≥55.8 μg/mL demonstrated significantly worse CFS compared with those with <55.8 μg/mL (median CFS, 136.6 months vs. not reached; hazard ratio (HR), 14.79; 95% confidence interval (CI), 5.11–42.77, *p* < 0.001) (Figure 3A). Similarly, when examining TRACP-5b, patients exceeding the threshold of 400 mU/dL exhibited markedly poorer CFS than those below this value (median CFS, 136.6 months vs. not reached; HR, 8.69; 95% CI, 2.83–26.72, *p* < 0.001) (Figure 3B). The amalgamation of bone turnover markers has been reported to augment the accuracy as prognostic factors [11]. Patients with P1NP and TRACP-5b values exceeding the cutoff were designated as the poor prognosis group; otherwise, they belong to the favorable group. This classification was designated as the bone turnover category (BTC). The poor prognosis group had significantly poorer CFS than the favorable group (median CFS, 53.0 months vs. not reached; HR, 17.39, 95% CI, 3.57–84.73, *p* < 0.001) (Figure 3C).

In the univariate analysis, PSA ≥ 20 ng/mL (HR, 27.12; 95% CI, 8.12–90.58, *p* < 0.001), GS ≥ 8 (HR, 49.58; 95% CI, 6.71–366.24, *p* < 0.001), T stage ≥ T3 (HR, 13.32; 95% CI, 5.33–33.28, *p* < 0.001), M1 (HR, 37.73; 95% CI, 16.41–86.74, *p* < 0.001), and BTC poor (HR, 18.68; 95% CI, 8.49–41.09, *p* < 0.001) emerged as significant prognostic factors for CFS. In the multivariate analysis, GS ≥ 8 (HR, 11.08; 95% CI, 1.24–98.96, *p* = 0.031), M1 (HR, 4.57; 95% CI, 1.43–14.61, *p* = 0.010), and BTC poor (HR, 3.64; 95% CI, 1.38–9.62, *p* = 0.009) remained significant prognostic factors for CFS (Table 2).

Based on the multivariate analysis, we further classified patients into two groups using three identified risk factors (BTC poor, GS ≥ 8, M1) associated with CFS. By introducing a novel risk classification, the low-risk group was defined as those possessing either none or one of these factors, whereas the high-risk group encompassed individuals with two or three factors. This risk classification was named the BGM classification after the initial letters of the three factors cited. The Kaplan–Meier survival curves for CFS are depicted in Figure 4. The median CFS was not reached in the low-risk group, whereas it was 19.1 months in the high-risk group (HR, 32.23; 95% CI, 6.80–152.8, *p* < 0.001). Thus, BGM risk classification using three factors indicates a much better HR than the single use of each factor, resulting in finding better discrimination for patients with poor prognoses. The BGM risk classification was also applied to evaluate CSS and OS. The median CSS was 85.3 months in the high-risk group, whereas it was not reached in the low-risk group (HR, 30.20; 95% CI, 4.89–186.4, *p* < 0.001) (Appendix A). The median OS was 179.7 months in the low-risk group compared with 81.6 months in the high-risk group (HR, 6.98; 95% CI, 2.24–21.72, *p* < 0.001) (Appendix A). These results demonstrate the potential utility of the BGM risk classification in predicting both CSS and OS.

### 3.3. Subgroup Analysis of Patients with ADT

A subgroup analysis was performed on 89 patients treated with ADT alone. The patient background is shown in Appendix A. The median follow-up period was 85.3 months. The median age at the time of PC diagnosis was 75 (range, 46–89) years, with a median PSA level of 17.3 (range, 1.5–16,701) ng/mL. Among them, 50 individuals (56%) had a GS of 8–10, 34 (38%) exhibited a clinical T stage of ≥3, 23 (16%) presented with lymph node metastases, and 25 (18%) had distant metastases. The median P1NP and TRACP-5b levels were 39.6 (range, 14.7–511) μg/L and 323 mU/dL (range, 104–1140 mU/L), respectively.

Patients with P1NP levels ≥ 55.8 μg/mL demonstrated significantly worse CFS compared with those with levels < 55.8 μg/mL (median CFS, 32.5 months vs. not reached; HR, 9.17; 95% CI, 3.70–22.74, *p* < 0.001) (Figure 5A). Similarly, patients with TRACP-5b levels ≥ 400 mU/dL demonstrated significantly worse CFS compared with those with levels < 400 mU/dL (median CFS, 63.1 months vs. not reached; HR, 4.98; 95% CI, 1.99–12.45, *p* < 0.001) (Figure 5B). Patients categorized as BTC-poor had significantly worse CFS compared with those categorized as BTC-favorable (median CFS, 19.14 months vs. not reached; HR, 7.63; 95% CI, 2.57–22.69, *p* < 0.001) (Figure 5C). Furthermore, the high-risk group in the BGM classification demonstrated significantly worse CFS compared with the low-risk group (median CFS, 17.1 months vs. not reached; HR, 14.31; 95% CI, 5.16–39.67, *p* < 0.001) (Figure 5D).

## 4. Discussion

This study identified three factors as predictors of CRPC: GS, presence of metastases, and high bone turnover markers. A multivariate analysis showed limited predictive power of PSA for castration resistance. PSA is a valuable biomarker for PC detection and risk stratification; however, its effectiveness as a prognostic factor has been questioned [12,13]. Both low and extremely high PSA levels also have poor prognostic value: PSA levels < 3.5 ng/mL result in more advanced cancer stage than patients with PSA levels between 3.5 and 10 ng/mL, and PSA levels > 100 ng/mL result in disproportionate overall or PC-specific survival rates [13]. Therefore, an alternative marker to PSA is needed at the start of treatment. In this study, the bone turnover marker showed no strong correlation with PSA, suggesting that it may be an alternative marker to PSA.

Bone turnover is the outcome of antagonistic activities between osteoblasts and osteoclasts. P1NP is generated during procollagen synthesis by osteoblasts, which cleaves at both the N- and C-terminal ends. This renders it useful as a marker for bone formation. On the contrary, TRACP-5b is an enzyme found exclusively in osteoclasts. It is released into the bloodstream because of increased bone resorption, thus serving as a marker for bone absorption [6]. Recent research has studied the measurement of P1NP and TRACP-5b as indicators of bone metastasis in PC. P1NP levels are significantly increased in patients with PC and bone metastasis [14,15]. Tracking P1NP variations has been reported to allow for the diagnosis of bone metastasis approximately 8 months before confirmation with bone scintigraphy [14]. Furthermore, the utility of P1NP in predicting responsiveness to treatments such as Ra-223 has been reported [16], solidifying its status as a noteworthy bone formation marker. Similarly, Yamamichi et al. developed a predictive model for bone metastasis in PC by combining TRACP-5b and PSA. The results demonstrated exceptional sensitivity (96.4%), specificity (80.3%), and an impressive AUC of 0.95. Additionally, this predictive model correlated significantly with clinical outcomes in CSS (*p* < 0.05) [17].

Previous studies have reported the clinical utility of bone turnover markers associated with bone metastasis [10,11], whereas the present study is the first to report that both P1NP and TRACP-5b, regardless of the presence of metastasis, are high in patients at risk of progression to CRPC. These results may be related to the interaction between bone and tumor cells, a phenomenon known as osteomimicry, wherein tumor cells acquire new characteristics [18,19,20].

Chemokines play a crucial role in this context. Chemokine CC motif ligand (CCL) 2 is a well-established prognostic factor in PC [3,21] and has received considerable attention in the complex tumor microenvironment. Suppression of androgen receptor (AR) in PC cell lines increases the migratory potential of PC cell lines via the CCL2/C-C chemokine receptor type 6-STAT3 axis [22]. CCL2 is also considered an important factor in osteoclast differentiation [23], and CCL2 deficiency reduced TRACP 5b levels [24]. Cocultivation of LNCaP PC cell lines and bone metastatic stromal cells enhanced the migratory ability of LNCaP via CCL5 [25]. In addition, CCL20 secretion by suppressed AR signaling in PC cells enhances migratory ability, activates fibroblasts, and induces osteoclasts, ultimately leading to bone resorption [26,27]. Bone turnover is facilitated by chemokine C-X-C motif ligand 12, a factor constitutively expressed by osteoblasts, and its receptors C-X-C chemokine receptor type (CXCR) 4 and CXCR6, thereby promoting bone metastasis [28,29].

In addition, interleukin-1β (IL-1β) is a notable factor, possessing dual capabilities of activating osteoclasts and enhancing the bone metastatic ability of PC cell lines. Moreover, IL-1β is highly expressed in PC tissues with a GS of ≥7, suggesting a potential for higher malignancy to activate bone turnover [30,31].

From the reports above, two hypotheses can be considered to explain the identification of P1NP and TRACP-5b as prognostic factors for CRPC. The first hypothesis posits that highly malignant PC releases an abundance of signals that activate osteoclasts and osteoblasts, leading to bone turnover activation. As a result, P1NP and TRACP-5b levels likely increase. The second hypothesis suggests that the activation of osteoclasts and osteoblasts may lead to the secretion of signals contributing to the acquisition of castration resistance. These two factors may synergistically accelerate cancer progression.

In this study, the finding that bone turnover markers could predict prognosis even in patients without bone metastases was expected because they may have acutely detected the presence of micrometastases, which CT and bone scintigraphy could not detect. Bone scintigraphy is commonly used to search for bone metastases; however, its sensitivity and specificity have been unsatisfactory at 95% and 80%, respectively. With the advent of whole-body MRI, its sensitivity and specificity have been significantly increased to 99% and 94%, respectively. Although the rise in whole-body MRI was a remarkable development, it still did not boast sufficient accuracy [32]. Another imaging modality of interest is gallium-68 prostate-specific membrane antigen positron emission tomography (PSMA-PET), which has a sensitivity and specificity of 97% and 77%, respectively, for detecting lymph nodes, far exceeding the sensitivity and specificity of simple CT, which are 42% and 82%, respectively. However, while the role of 68Ga-PSMA-PET in detecting lymph node metastases is well established, its exact utility in detecting bone metastases has yet to be discussed because of insufficient high-level data [33]. Although the accuracy of the test is increasing, imaging alone has limitations, and it may be possible to supplement its detection power with bone turnover markers.

The limitations of this study include its retrospective design, single-center focus, and relatively small sample size. In addition, potential interactions with other medications have yet to be explored.

Uncertainties regarding the optimal cutoff values for P1NP and TRACP-5b must also be considered. In this study, we did not use the P1NP and TRACP-5b cutoff values at the upper limit of normal on the attached text because the cutoff values in the accompanying text include young people in their 20s, which deviates from the median age of the PC population; thus, we considered them only as a reference. In addition, in other studies, the cutoff values for P1NP and TRACP-5b were set at 30.3–75 μg/L [10,34,35] and 353 mU/dL [17], respectively, and the values of 55.8 μg/L for P1NP and 400 mU/dL for TRACP-5b set in the present study were not significantly different from those in previous reports. Collecting more cases to derive accurate cutoff values is necessary for full-scale clinical applications. Owing to the limited number of cases, stratification based on treatment strategy was statistically underpowered. However, the analysis of cases in which only ADT was performed as a subgroup showed the usefulness of the BGM risk classification and that of the overall population, indicating the significance of revisiting this issue with a more significant number of cases in the future. The limited number of mortality events posed challenges in conducting a comprehensive multivariate analysis for CSS and OS predictors. Nevertheless, considering that PC inevitably progresses to castration resistance before leading to PC-related death, CFS was regarded as a primary indicator for both CSS and OS. Indeed, the BGM risk classification, anchored in CFS, demonstrated its efficacy as a reliable predictor for both CSS and OS (Appendix A). We had a relatively long observation period, with a median observation period of 115.1 months. However, prostate cancer often has a relatively good prognosis and requires long-term follow-up [2]. Since this study examined prognostic prediction, longer follow-up was necessary, which was also a limitation of this study.

## 5. Conclusions

This study establishes that the serum markers P1NP and TRACP-5b hold promise as prognostic factors for CRPC. The findings suggest that these bone turnover markers offer valuable insights into disease progression and survival outcomes. While acknowledging this study’s limitations, the promising results suggest a hopeful direction for future research. The utility of P1NP and TRACP-5b holds promise for enhancing diagnostic and prognostic strategies in the context of PC.

## Figures and Tables

**Figure 1 biomedicines-12-00292-f001:**
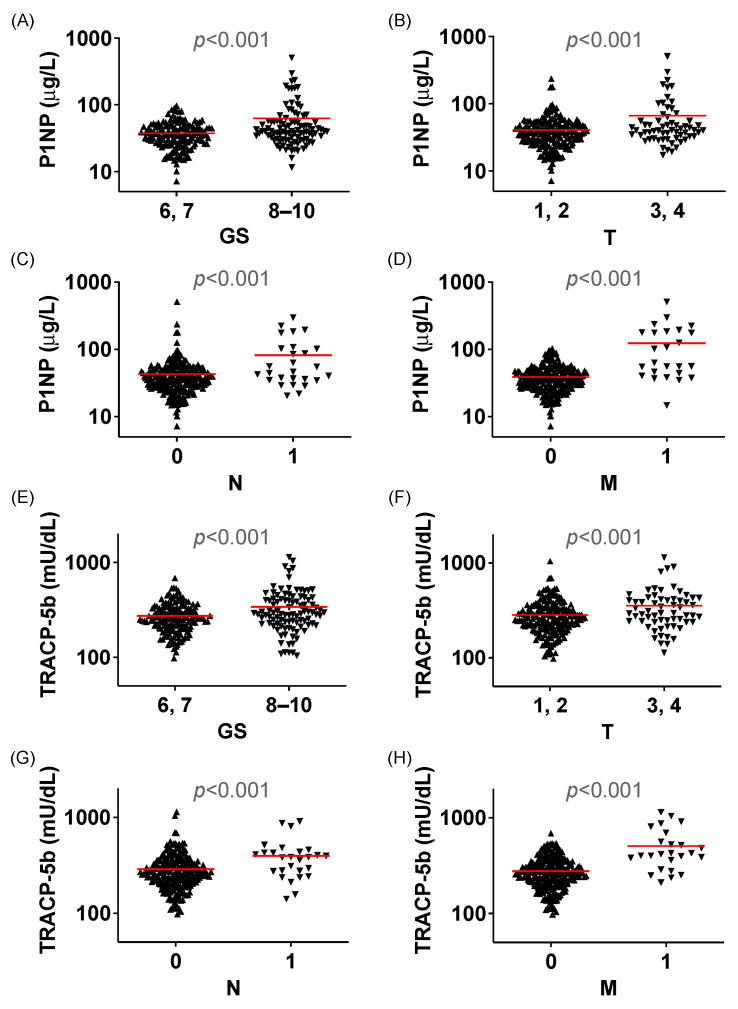
P1NP values for individual patients were divided per the Gleason score (**A**), T classification (**B**), N classification (**C**), and M classification (**D**). TRACP-5b values for individual patients were divided per the Gleason score (**E**), T classification (**F**), N classification (**G**), and M classification (**H**). The bar represents median values. P1NP, peptides n-terminal propeptide of type I procollagen; TRACP-5b, tartrate-resistant acid phosphatase type 5b.

**Figure 2 biomedicines-12-00292-f002:**
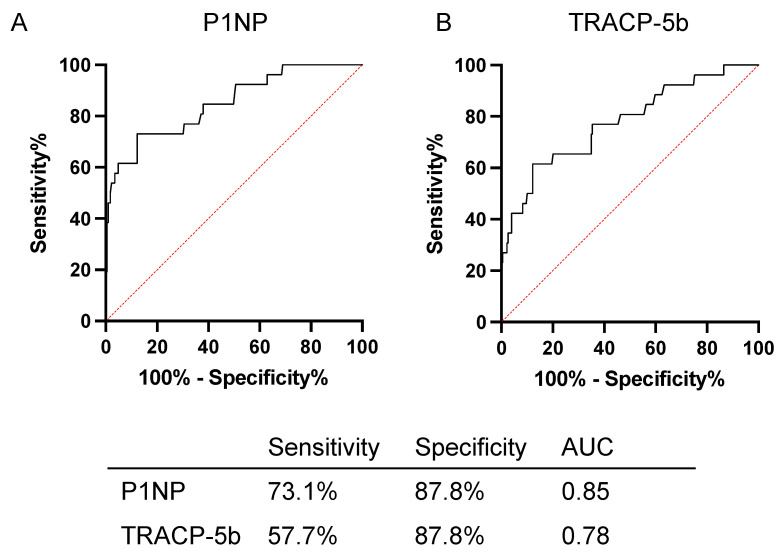
Receiver operating characteristic curves of the predicted probability of castration resistance by P1NP and TRACP 5b. The red line indicates the line of identity. AUC, area under the curve; P1NP, peptides n-terminal propeptide of type I procollagen; TRACP-5b, tartrate-resistant acid phosphatase type 5b.

**Figure 3 biomedicines-12-00292-f003:**
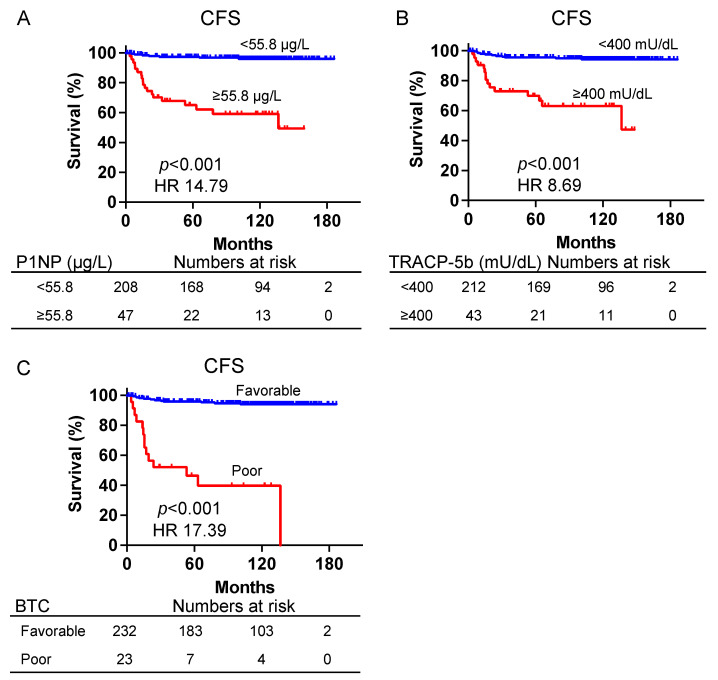
Kaplan–Meier curves for the CFS of patients with prostate cancer with P1NP ≥ 55.8 μg/L and P1NP < 55.8 μg/L (**A**), TRACP-5b ≥ 400 mU/dL and TRACP-5b < 400 mU/dL (**B**), and BTC favorable and poor (**C**). BTC: bone turnover category—designating those with both P1NP and TRACP-5b above the cutoff as “poor” and the remainder as “favorable”; CFS, castration-resistant prostate cancer-free survival; P1NP, peptides n-terminal propeptide of type I procollagen; TRACP-5b, tartrate-resistant acid phosphatase type 5b.

**Figure 4 biomedicines-12-00292-f004:**
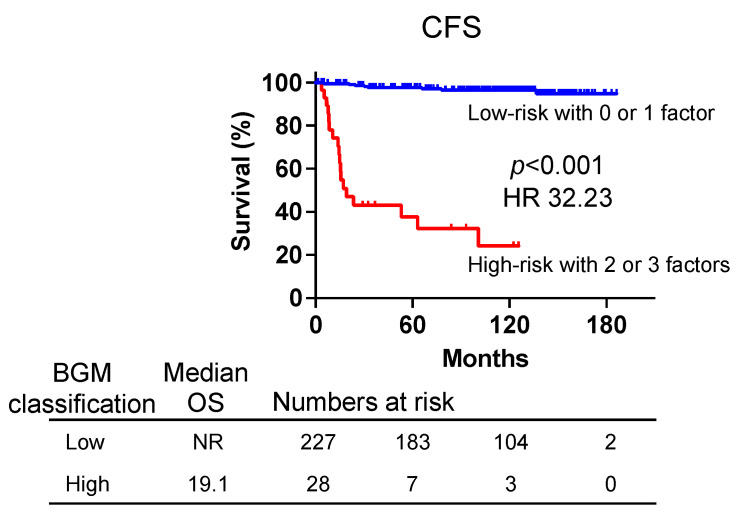
The BGM classification system was created by combining the poor bone turnover category, a Gleason score ≥ 8, and metastasis to predict patient outcomes. The low-risk group had none or one risk factor, and the high-risk group had two or three factors. Kaplan–Meier curves for the CFS of classified patients with PC as having low and high risk. CFS, castration-resistant PC-free survival.

**Figure 5 biomedicines-12-00292-f005:**
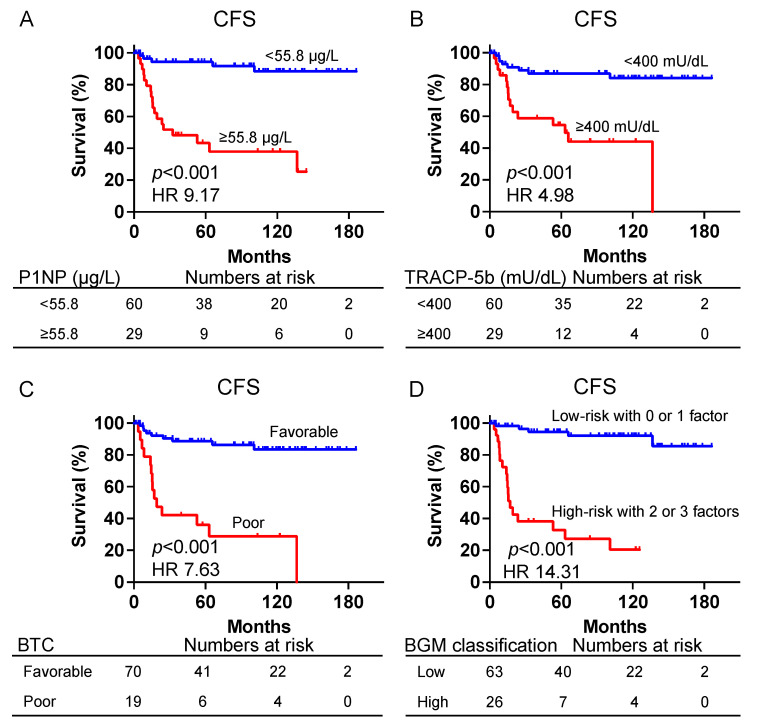
Kaplan–Meier curves for CFS of patients with prostate cancer treated only with androgen deprivation therapy with P1NP ≥ 55.8 and P1NP < 55.8 μg/L (**A**), TRACP-5b ≥ 400 and TRACP-5b < 400 mU/dL (**B**), and BTC favorable and poor. (**C**) BGM classification as low risk and high risk (**D**). BTC, bone turnover category; CFS, castration-resistant prostate cancer-free survival; P1NP, peptides n-terminal propeptide of type I procollagen; TRACP-5b, tartrate-resistant acid phosphatase type 5b.

**Table 1 biomedicines-12-00292-t001:** Patient characteristics.

Patients, *n*		255	
Median follow-up, month (range)		115.1	(1.3–186.2)
Median age, year (range)		69	(46–89)
Median PSA, ng/mL (range)		10.6	(1.5–16701)
Gleason score, *n* (%)	≤7	157	(62)
	≥8	98	(38)
T stage, *n* (%)	≤2	193	(76)
	≥3	62	(24)
N stage, *n* (%)	0	228	(89)
	1	27	(11)
M stage, *n* (%)	0	230	(90)
	1	25	(10)
Treatment, *n* (%)	ADT alone	89	(35)
	RT ± ADT	127	(50)
	RP ± ADT	35	(14)
	AS	4	(2)
Median P1NP, μg/L (range)	Median (range)	37.9	(7.2–511)
Median TRACP-5b, mU/dL (range)	Median (range)	273	(98–1140)
Death at data cutoff date, *n* (%)	All cause	49	(19)
	Prostate cancer specific	17	(7)
Development to CRPC, *n* (%)	Number	26	(10)

ADT, androgen deprivation therapy; AS, active surveillance; CRPC, castration-resistant prostate cancer; PSA, prostate-specific antigen; P1NP, peptides n-terminal propeptide of type I procollagen; RT, radiation therapy; RP, radical prostatectomy; TRACP-5b, tartrate-resistant acid phosphatase type 5b.

**Table 2 biomedicines-12-00292-t002:** Univariate and multivariate analyses of castration-resistant prostate cancer-free survival.

			Univariate			Multivariate		
		*n*	*p*-Value	HR	95% CI	*p*-Value	HR	95% CI
Age, years	<70	129	0.75	1.13	0.52–2.45			
	≥70	126						
PSA, ng/mL	<20	184	<0.001	27.12	8.12–90.58	0.093	3.82	0.80–18.28
	≥20	71						
GS	≤7	157	<0.001	49.58	6.71–366.24	0.031	11.08	1.24–98.96
	≥8	98						
T stage	≤T2	193	<0.001	13.32	5.33–33.28	0.92	1.06	0.33–3.42
	≥T3	62						
M stage	M0	230	<0.001	37.73	16.41–86.74	0.010	4.57	1.43–14.61
	M1	25						
BTC	Favorable	232	<0.001	18.68	8.49–41.09	0.009	3.64	1.38–9.62
	Poor	23						

BTC: bone turnover category, which is made of peptides n-terminal propeptide of type I procollagen and tartrate-resistant acid phosphatase type 5b. CI, confidence interval; GS, Gleason score; HR, hazard ratio; PSA, prostate-specific antigen.

## Data Availability

The data presented in this study are available within this article.

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
