# Peer review of "Bone Turnover Markers, n-Terminal Propeptide of Type I Procollagen and Tartrate-Resistant Acid Phosphatase Type 5b, for Predicting Castration Resistance in Prostate Cancer"

_biomedicines, 2024, doi:10.3390/biomedicines12020292_

Round 1

Reviewer 1 Report

Comments and Suggestions for Authors

Methods: 

  • Line 109: Why did the authors not include other relevant covariates into the multivariate survival analysis? Such as comorbidity status or type of treatment received?

  • How did authors choose which patients had P1NP and TRACP-5b levels ordered in this retrospective trial? These criteria must be disclosed given potential for selection bias.

  • Did authors include PSA nadir, or percentage change PSA, or PSA50 (proportion of patients whose PSA declined >50%)

  • The methodology may also be confounded by including patients at various stages, rather than focusing on either localized or metastatic patients, as well as de novo diagnosis vs. recurrent disease, and patients treated with surgery and radiation. 

  • It would be more meaningful, for example, to assess rates of biochemical recurrence after radical prostatectomy using traditional variables (PSA, path stage, Gleason grade, margin status) and to also include BTC to see if this is a separate significant prognostic variable. The authors do mention, however, that this type of stratification resulted in low numbers that confounded meaningful statistical analysis. 

  • Also the definition of CRPC as PSA >2ng/mL and 25% increase from nadir value MUST be in the context of testosterone level <20ng/dL (the accepted definition of “castration”), which indicates active androgen-deprivation therapy. For example, PSA of 2ng/mL after radical prostatectomy does not denote castration-resistance. I think the authors are just trying to assess cancer-free survival in general but are misattributing it as “castration-resistant” prostate cancer survival, which is not the same thing.

Results:

  • Line 194: Did the authors perform any assessment for collinearity between BTC status and M1 status. Given that P1NP and TRACP-5b were significantly elevated in M1 patients compared to M0 patients, it may be worth assessing the inclusion of both into the multivariate analysis.

  • Line 128: is this radiotherapy to the prostate or radiotherapy to metastatic sites?

  • *The very long median follow-up time is commendable. 

  • Measurement of these bone turnover markers at various time points would be more meaningful than a single time-point at the time of prostate biopsy. 

  • The ROC curve analysis is well-done

Limitations:

  • Line 338: Since the patients analyzed in this study were of Asian descent and from one institution, the generalizability of these results is limited as noted by the authors. Have prior studies on these markers in PC used cohorts of patients of different races?

Reviewer 2 Report

Comments and Suggestions for Authors

The authors presented that two bone turnover markers, n-terminal propeptide of type I procollagen (P1NP) and tartrate-resistant acid phosphatase type 5b 16 (TRACP-5b) are useful for detection of prognosis on prostate cancer. Additionally, the combination of P1NP and TRACP-5b levels in the serum, defined as having a poor bone turnover category (BTC), indicated more significantly separation for the prognosis of castration-resistant prostate cancer recurrence-free survival (CFS). These data are very interesting and useful for the prognosis of prostate cancer, but some questions and suggestions as described below.

Major comments

1- The authors presented that “a weak correlation was found between TRACP-5b and age” and “a weak correlation was observed between P1NP and PSA” on page 4, lines 144 to 145. However, their R2 is less than 0.1. In general, if two categories have a correlation between -0.1 and 0.1, it is assumed that there is no relationship between them. They should correct them.

2- How about the correlation between P1NP and TRACP-5b values in the serum of prostate cancer patients? Are there common regulatory and/or regulated factors for them?

3- What does each table in Figure 3, 4 and 5 show? It seems that the data in the table and the graph do not match. The “Numbers at risk” data does not reflect the percentage of survival data in the graph.

Minor comments

1- The authors should present the formal name of “P1NP/P1CP” on page 1, line 45.

2- In Supplementary table1, the explanation of abbreviations such as “RT, radiation therapy; RP, radical prostatectomy” is not required.

Reviewer 3 Report

Comments and Suggestions for Authors

The authors in this study retrospectively examined the usefulness of P1NP and TRACP-5b as prognostic biomarkers. The authors should be commended on this single institution effort. 

The study is methodologically well performed.

There are a few of points which may be considered for further improvement.

The mean follow-up time could be longer, given that it is a cohort retrospective study, i.e. examination of prognostic factors. Which is one of the main limitations of this study. State in limitations a short follow-up time.

Please provide reference justifying Previous studies have reported the clinical utility of bone turnover markers associated with bone metastasis...” Section 4

“A 10-core needle transrectal ultrasound-guided biopsy of the prostate gland”, is that the standard of care in your country?

The findings of this study are significant for clinical practice.

Round 2

Reviewer 2 Report

Comments and Suggestions for Authors

Authors ameliorated the manuscript with reviewers’ comments.